# Proteome and phosphoproteome signatures of recurrence for HPV$^+$ head and neck squamous cell carcinoma

Tomonori Kaneko [1], Peter Y. F. Zeng[2,3,4], Xuguang Liu[1], Rober Abdo[3], John W. Barrett[2], Qi Zhang[3], Anthony C. Nichols [2,4✉] & Shawn Shun-Cheng Li [1✉]

## Abstract

**Background** Head and neck squamous cell carcinoma (HNSCC) is the sixth most common cancer worldwide and the human papillomavirus (HPV$^+$)-driven subtype is the fastest rising cancer in North America. Although most cases of HPV$^+$ HNSCC respond favorably to the treatment via surgery followed by radiochemotherapy, up to 20% recur with a poor prognosis. The molecular and cellular mechanisms of recurrence are not fully understood.

**Methods** To gain insights into the mechanisms of recurrence and to inform patient stratification and personalized treatment, we compared the proteome and phosphoproteome of recurrent and non-recurrent tumors by quantitative mass spectrometry.

**Results** We observe significant differences between the recurrent and non-recurrent tumors in cellular composition, function, and signaling. The recurrent tumors are characterized by a pro-fibrotic and immunosuppressive tumor microenvironment (TME) featuring markedly more abundant cancer-associated fibroblasts, extracellular matrix (ECM), neutrophils, and suppressive myeloid cells. Defective T cell function and increased epithelial-mesenchymal transition potential are also associated with recurrence. These cellular changes in the TME are accompanied by reprogramming of the kinome and the signaling networks that regulate the ECM, cytoskeletal reorganization, cell adhesion, neutrophil function, and coagulation.

**Conclusions** In addition to providing systems-level insights into the molecular basis of recurrence, our work identifies numerous mechanism-based, candidate biomarkers and therapeutic targets that may aid future endeavors to develop prognostic biomarkers and precision-targeted treatment for recurrent HPV$^+$ HNSCC.

### Plain language summary

Head and neck cancer can be caused by the human papillomavirus. While this type of cancer responds well to chemotherapy given simultaneously with radiation, a significant proportion of cases recur within a few years, leading to illness and sometimes death in these patients. It is therefore important to understand the mechanisms of recurrence in order to develop better treatments. By comparing the levels of proteins and protein phosphorylation—a type of modification that affects how proteins work—between tumors from patients with or without recurrence, we found that the cells surrounding recurrent tumors show signs of fibrosis—the development of fibrous connective tissue—and suppression of the body's immune responses. This suggests that therapies directed towards the regulators of fibrosis and immune suppression may help to overcome recurrent head and neck cancer.

[1] Department of Biochemistry, Schulich School of Medicine and Dentistry, Western University, London, ON N6A 5C1, Canada. [2] Department of Otolaryngology - Head and Neck Surgery, Schulich School of Medicine and Dentistry, Western University, London, ON N6A 5C1, Canada. [3] Department of Pathology and Laboratory Medicine, Schulich School of Medicine and Dentistry, Western University, London, ON N6A 5C1, Canada. [4] Lawson Research Institute, 268 Grosvenor St, London, ON N6A 4V2, Canada. ✉email: anthony.nichols@lhsc.on.ca; sli@uwo.ca

HPV infection causes approximately 25% of all HNSCC cases and the rate has been rising in recent years[1]. HPV+ HNSCC is biologically and clinically distinct from non-HPV driven (HPV-) HNSCC, which is typically associated with tobacco and alcohol consumption[2–5]. Despite the availability of an HPV vaccine, modelling using current vaccination rates suggests a continued rise in cases until at least 2045[6], indicating that HPV+ HNSCC will be a clinical challenge for the foreseeable future. The most common treatment for HPV+ HNSCC is surgery and high-dose cisplatin chemotherapy given concurrently with radiation[7,8]. While this treatment strategy cures the majority of patients[9,10], significant long-term patient burdens can occur, including difficulty swallowing, kidney damage, osteoradionecrosis, and delayed death due to chronic aspiration[11]. Because many of these patients with HPV+ HNSCC will survive, they must cope with the toxicity of their treatment potentially for decades. Conversely, as the number of patients continue to rise, encountering patients with relapsed disease is becoming an increasingly common clinical scenario. Thus, there is an unmet need to develop accurate biomarkers to de-intensify treatment for HPV+ HNSCC with low risk of relapse. In the meantime, it is important to develop novel and more effective therapies for patients with recurrent HNSCC because of their generally poor response to conventional treatment regimens.

The mechanism of HNSCC recurrence is not fully understood to date. Emerging evidence suggests that the amount, type, and levels of activation of tumor-infiltrating lymphocytes are associated with response to conventional therapy and immunotherapy or survival[12–18]. For example, a recent multi-omics study carried out on HPV− HNSCC identified actin dysregulation and immunosuppression, due to widespread deletion of immune regulatory genes, as important determinants of disease pathology and response to treatment[19]. Intriguingly, these insights were obtained primarily from analysis of the tumor proteome rather than the transcriptome as the protein data substantially outperformed RNA data in co-expression-based function prediction[19]. This is reminiscent of another recent study demonstrating that the proteome, but not the transcriptome, of pancreatic ductal adenocarcinoma (PDAC), was able to differentiate different tumor groups[20]. Furthermore, large-scale proteogenomic studies estimates that only 20% of protein-coding genes have high correlations between the transcript and protein abundance[21]. The poor-to-moderate correlation between the transcriptome and proteome data underscores the importance of characterizing the tumor proteome independently or together with the transcriptome. In this regard, we note that no systematic proteomic study has been carried out to date on HPV+ HNSCC and consequently, the proteome of this cancer subtype remains unexplored.

Here we report a deep and quantitative mass spectrometry (MS) analysis of 15 HPV+ HNSCC samples, including 7 with recurrence. Our study uncovered systematic changes in the proteome and phosphoproteome between the recurrent and non-recurrence tumors that suggests extensive remodeling of the tumor microenvironment and reprogramming of the kinome associated with recurrence. Furthermore, numerous mechanism-based biomarkers and potential therapeutic targets emerged from our deep proteomic and phosphoproteomic analysis, yielding a valuable resource for future exploration.

## Methods

**Patient cohort**. Fresh tumor samples were prospectively collected from patients with HPV+ oropharyngeal squamous cell cancer at the Victoria Hospital and London Health Science Centre, London, Ontario, Canada between 2010 and 2016. The study was approved by the Research Ethics Board at Western University (REB 7182) and informed written consent was obtained from each patient. The samples were frozen immediately after surgical resection using the optimal cutting temperature compound as cryo embedding matrix. Patient demographics and survival outcomes were prospectively collected. Frozen section analysis was carried out to confirm tumor cellularity greater than 70%. HPV status was confirmed via PCR and Sanger sequencing. All the primary tumor samples (NR1-NR8, and R1-R3) were collected before any treatment. For the samples R4-R7, the patients received cisplatin or radiation treatment until no evidence of disease was observed. However, their tumors redeveloped after the treatment was discontinued. The R4-R7 relapse tumor samples were collected by salvage surgery. None of the 15 patients received immunotherapy. Detailed clinical information is provided in the Supplementary Table 1.

**Sample processing for mass spectrometry analysis**

*Tissue processing*. The frozen tissue was washed in ice-cold phosphate-buffered saline (PBS) once and homogenized in a precooled mortar and pestle, then resuspended in 1 ml cold buffer containing 8 M urea, 50 mM Tris-HCl (pH 7.6), 2% (v/v) proteinase inhibitor (Sigma, P8340) and 1 mM NaF. The mixture was rotating-mixed at 4 °C for 20 min to dissolve the proteins. Tissue debris was removed by centrifugation, and the supernatant was transferred into a fresh tube and mixed with 5-fold volume of cold precipitation mixture (containing 50% acetone, 50% ethanol, and 0.1% acetic acid) and incubated overnight at −20 °C.

*Cell culture and pervanadate treatment for the booster channel*. The patient-derived HNSCC cell line 93VU147T was tested for mycoplasma contamination and kept in Dulbecco's Modified Eagle Medium (DMEM) supplemented with 10% FBS and 1% penicillin-streptomycin. T cells were isolated from blood of healthy donors. Both 83VU137T cells and T cells were treated with sodium pervanadate as follows. Briefly, the pervanadate solution was prepared by adding 10 μl of 0.1 M sodium orthovanadate to 10 μl of 0.2 M hydrogen peroxide (diluted 50-fold from a 30% stock). The solution was then incubated at room temperature for 15 min. Excess hydrogen peroxide was inactivated by adding 2 μl of catalase in PBS (10 mg/ml).

*Protein processing and digestion*. The protein precipitate was collected by centrifugation, washed with 75% ethanol, and then redissolved in 200 μl 9 M urea containing 50 mM HEPES, pH 8.0, and 10 mM DTT. The mixture was incubated for 1 h at room temperature. Subsequently, the sample was alkylated with 28 mM iodoacetamide for 40 min in the dark and the reaction was quenched by adding 10 mM DTT. The sample was diluted by adding 700 μl 50 mM HEPES, pH 8.0, 1 mM sodium orthovanadate, and digested with LysC (1 mAU per 25 μg protein) for 2 h at 28 °C, followed by incubating with trypsin at 1:50 enzyme-to-substrate ratio overnight at 28 °C. The sample was acidified by adding TFA to 0.5% final concentration and was desalted on a C18 SPE column (Waters WAT054955). The peptides were eluted in 70% acetonitrile/0.1% formic acid and dried by Speedvac.

*Tandem Mass Tag (TMT) labelling*. For mass spectrometry analysis, we labelled 15 patient and 5 healthy control samples with the 11-plex TMT isobaric labelling reagent (Thermo A37725; see Supplementary Table 2 for sample identities with TMT set/channel numbers). In addition, we employed the pervanadate BOOST channel approach[22] by including a 1:1 mixture of the pervanadate-treated 93VU147T cells and T cells as the reference channel (channel 1 of each 11-plex sample). Three sets of 11-plex reagents were used to label all samples.

The TMT labelling procedure was modified from Chua et al.[22]. Briefly, the desalted peptides were reconstituted in 0.1% formic acid to determine peptide concentration by the BCA protein assay kit (Pierce 23225). Peptides in 180 µg portions from each sample were aliquoted and vacuum-dried. A reference sample was prepared by pooling the samples C3, NR1, NR3, and NR6. Each of 0.8 mg 11-plex TMT labelling reagents was reconstituted in 41 µl acetonitrile. The peptides were reconstituted in 36 µl of 0.1 M HEPES (pH 8.5) to prepare 5 mg/ml peptide solution and were mixed with 13.5 µl of the TMT reagent to label the peptides at room temperature for two hours. After the reaction, a 1 µl aliquot was taken from each sample to check TMT labelling efficiency by mass spectrometry. The reaction was quenched by adding 2.7 µl of 5% hydroxylamine. The 11 samples in a TMT set were combined (2 mg total peptides) and desalted on SepPak C18 cartridges.

*Phosphopeptide enrichment.* We previously reported that engineered Src SH2 domains called the SH2 superbinder (SH2S) could be used to enrich pTyr peptides for mass spectrometry analysis[23]. SH2S-Agarose beads (Precision Proteomics, London, Canada) were used for pTyr peptide enrichment. Briefly, the TMT-labelled peptides were reconstituted in 50 mM ammonium bicarbonate and incubated with 100 µl SH2S beads for 30 min at room temperature with rotation. The flow-through fraction was saved for pSer/Thr enrichment by $Ti^{4+}$-immobilized metal affinity chromatography (IMAC)[24] (a gift from Dr. Mingliang Ye, Dalian Institute of Chemical Physics). The beads were washed twice in 0.2 M ammonium bicarbonate, followed by 2x washes in 50 mM ammonium bicarbonate. The bound pTyr peptides were eluted using 0.4% trifluoroacetic acid (TFA). The eluted pTyr peptides were directly loaded onto the fractionation columns (see next section).

The flow-through fraction contains peptides not captured by the SH2S-agarose beads. Serine/threonine-phosphorylated peptides were enriched by $Ti^{4+}$-IMAC resin following a published protocol[24]. Briefly, a 500 µg portion of the flow-through fraction from the SH2S enrichment step was mixed 1:1 (v/v) with 80% acetonitrile/6% TFA solution and then loaded to the IMAC resin. After incubation and wash steps (wash-1 solution: 50% acetonitrile, 6% TFA, 200 mM NaCl, wash-2 solution: 30% acetonitrile, 0.1% TFA), the peptides were eluted by 10% ammonia, and dried by Speedvac.

*High-pH fractionation.* The peptides were separated by Pierce high pH reversed-phase peptide fractionation kit (Thermo 84868). For proteome analysis, 10-µg portions of the peptides after TMT labelling were diluted in 300 µl 0.1% TFA for loading and 8 fractions (10–50% acetonitrile in 0.1% triethylamine) were collected. The IMAC-enriched peptides were resuspended in 300 µl 0.1% TFA and loaded onto the fractionation column. The SH2S-enriched peptides, upon elution in 0.4% TFA from the SH2S-agarose beads, were loaded directly to the fractionation column. For the IMAC and SH2S-enriched peptides, the peptides were eluted at 5, 10, 12.5, 15.0, 17.5, 20.0, 22.5 and 50% acetonitrile in 0.1% triethylamine in a step-wise manner and the resulting fractions (8) were concatenated into 4 tubes (i.e., combining fractions 1-5, 2-6, 3-7 and 4-8). The fractionated peptides were dried on Speedvac.

*LC-MS/MS experiments.* The fractionated peptides were reconstituted in 2% acetonitrile/0.1% formic acid (FA). The peptides were analyzed by the data-dependent acquisition (DDA) method on an EASY-nLC 1000 system coupled to a Q-Exactive Plus mass spectrometer (Thermo Fisher Scientific). The peptides were loaded on an Acclaim PepMap 100 C18 column (20 mm with 75 µm in diameter, Thermo 164946), and separated on an EASY-Spray ES803A analytical column (Thermo Fisher Scientific) at the flow rate of 300 nl/min and a linear gradient from 3 to 40% acetonitrile in 0.1% formic acid. The gradient length was 2 hours for the pTyr phosphoproteome fractions and 4 h for the proteome or the IMAC phosphoproteome fractions. See Supplementary Table 3 for parameters of mass spectrometry data acquisition.

**MS data processing.** FragPipe version 16.0 was used for data processing[25]. For the proteome and IMAC datasets, the pool channel was used as the reference channel for batch correction. Due to the reporter ion interference from the phosphotyrosine boost channel (channel 1) to the pool channel (channel 3) caused by isotopic impurities of the TMT reagents, the pool channel was not usable for TMT batch correction for the SH2S-enriched pTyr datasets. Instead, the technical triplicate sample (the tonsil control C2 in all three TMT batches, see Supplementary Table 2) was used as the reference channels for batch correction. The human protein sequences obtained from UniProt (20367 reviewed entries, February 2020) were used in the database search. For the proteome data processing, the TMT10-bridge workflow was used for the proteome search (by changing to the plex to TMT11). For the phosphoproteome data, the TMT10-phospho-bridge workflow was loaded, followed by changing the plex to TMT11 and the minimal peptide length to 6. The median centering normalization by FragPipe was applied to the log2 intensities.

**Proteome and phosphoproteome data analysis.** For data analysis, only the proteins (for proteome) or phosphosites (for phosphoproteome) observed in at least three samples in each group (control, non-recurrent (NR) or recurrent (R)) were retained. Perseus version 1.6.14.0 was used to analyze the data[26]. The VolcaNoseR server was used for drawing volcano plots. The list of the human kinases was based on Manning et al.[27]. The list of meta signatures that reflect common expression programs are taken from Puram et al.[28]. The Metascape server was used for functional gene annotation analysis[29]. Pathview Web was used for KEGG pathway analysis[30]. The heatmaps were prepared with the Morpheus server. The master protein-protein interaction network was constructed by extracting proteins and phosphosites significantly more abundant in recurrence samples (>1.5-fold more abundant in the recurrence group, and $p < 0.1$ between NR and R groups) in the mass spectrometry data. The STRING database[31] was used to connect the nodes on Cytoscape (confidence cutoff 0.9)[32]. For the Kaplan–Meier survival analysis, the data for 71 HPV+ HNSCC cases were retrieved from The Cancer Genome Atlas (TCGA). For each gene, the 71 patients were divided into high expression ($n = 36$) or low expression group ($n = 35$), and the disease-specific survival month was plotted using GraphPad Prism 9.

**Statistical analysis.** The log2 intensity values were used for statistical analysis in volcano plots and box plots. Unpaired 2-sided t-tests were conducted between the non-recurrence and recurrence groups using Perseus. GraphPad Prism 9 was used for the survival analysis, with the log-rank test. The box in the box plot extends from the 25th to 75th percentiles. The center line is plotted at the median. The whiskers go down to the smallest value and up to the largest.

**Reporting summary.** Further information on research design is available in the Nature Research Reporting Summary linked to this article.

## Results

### The proteome and phosphoproteome are significantly different between recurrent and non-recurrent tumors.

The 15 tumor samples used in the MS analysis included those from 8 patients that were tumor-free and 7 with recurring tumors. The 7 recurrent samples (R) contained 3 surgically resected primary tumors (R1-R3) with future recurrence and 4 secondary tumors collected from salvage surgery after relapse (R4-R7) (Supplementary Table 1). We also included 5 normal tonsil tissue samples (C) as controls in the MS analysis. The MS samples were divided randomly into three groups for tandem mass tag (TMT) labelling (Supplementary Table 2). Each batch of TMT-11plex also contained a mixture of peptides from the HNSCC patient-derived cell line 93VU147T and T cells (upon treatment with sodium pervanadate) in channel 1 to boost detection of tyrosine-phosphorylated peptides[22], and a pool of tumor samples (in channel 3) for batch-to-batch normalization[33]. To facilitate phosphoproteomics analysis, we employed the $Ti^{4+}$-IMAC beads to enrich pSer/pThr-containing peptides and the SH2 Superbinder (SH2S) beads to capture pTyr-containing peptides prior to identification by nano liquid chromatography (nanoLC)-tandem MS (MS/MS)[23] (Supplementary Table 3). Collectively, our MS analysis identified 7288 proteins and 10,423 phosphosites, including 9253 pSer/pThr and 1170 pTyr sites (Fig. 1a) that were detected in at least 3 samples in both the patient and control groups. To the best of our knowledge, this represents the largest proteome and phosphoproteome datasets reported for HPV[+] HNSCC to date.

We used uniform manifold approximation and projection (UMAP) analysis to reduce the dimensionality of the proteome data and determine whether the recurrent (R) and non-recurrent (NR) tumor groups could be distinguished. We found that the proteome was segregated according to recurrent status (Fig. 1b). Intriguingly, the primary tumors with future recurrence (R1-R3) were located between the recurrence-free primary tumors and the relapsed, secondary tumors on the UMAP plot. This suggests that the primary tumors with future recurrence acquired certain characteristics of the recurrent tumors prior to disease relapse.

Volcano plots were used to identify the significantly altered proteins and phosphosites between groups, followed by Gene Ontology (GO) analysis to identify the associated biological processes and/or functions. Of the 307 proteins exhibiting significant changes with recurrence, 284 were found increased and 23 decreased in the recurrent group. The proteins with the greatest increases in the recurrent tumors included the neutrophil collagenase MMP-8, neutrophil defensin 3 (DEFA3), neutrophil elastase (ELANE), matrix metalloproteinase-9 (MMP-9), collagens, fibrinogens (FGG, FGB), and components of the complement system or of thrombosis (Fig. 1c; Supplementary Table 4). In contrast, the most decreased proteins in the recurrent group included CD3γ, CD8A, CD2, and CD6, which play essential roles in T cell receptor (TCR) signaling and/or T cell interaction with other cells (Fig. 1c; Supplementary Table 5). This suggests that decreased T cell infiltration into the tumor is associated with the recurrence. GO analysis of the differentially expressed proteins (DEPs) further identified extracellular matrix (ECM) organization/regulation, neutrophil degranulation, and complement and coagulation as the most significantly up-regulated processes/functions associated with recurrence (Fig. 1d). Comparison of the phosphoproteome data identified 157 significantly increased and 109 significantly decreased phosphosites in the recurrence relative to the no-recurrence group (Fig. 1e; Supplementary Table 6). The phosphorylated proteins/phosphosites exhibiting the largest increases in the recurrence group included FGG-pY448, STAT5A/B-pY694/669, and SIT1-pY188. In contrast, MYH2-Y1381, CD8A-S231, CD247 (CD3ζ)-Y111, CD3G-Y160, LCK-Y505, and VAV1-Y826 were among those displaying the largest decreases in the recurrent tumors (Fig. 1e; Supplementary Table 6). STAT5 phosphorylation (activation) is known to suppress anti-tumor immunity and promote tumor cell proliferation, invasion and survival[34]. SIT1 is a negative regulator whereas CD8A, CD3, LCK, VAV1 are positive regulators of TCR signaling. The significant differences in phosphorylation for these TCR regulators are consistent with T cell and TCR dysfunction in recurrent tumors. On the other hand, the increased phosphorylation of FGG is consistent with elevated fibrosis potential, cytoskeletal reorganization, and cell migration—GO processes that were found specifically enriched in the recurrence group (Fig. 1f). Collectively, the differential protein expression and phosphorylation analysis suggest that the recurrent tumors are characterized with an enriched ECM, an increased metastatic potential, and a decrease in T cell-mediated anti-tumor immune response.

### Recurrence is driven by TME remodeling and increased EMT potential.

Because the tumor microenvironment (TME) plays a critical role in tumor progression, we next analyzed the proteome using the knowledge-based functional gene expression signatures (Fges) that represent the major functional and cellular components of the tumor and TME[35]. Remarkable differences in specific Fges were found not only between the tumor and tonsil control groups, but also between the no-recurrence and recurrence groups (Fig. 2a). Of note, a significant reduction in the protein levels of CD8A, CD3G, CD3D, CD5, ZAP70, GZMA and GZMK, respectively, were detected in the recurrent group. Together with the results from the DEP analysis (Fig. 1e), this suggests that both the tumor-infiltration and the effector function of cytotoxic (CD8[+]) T cells were compromised in the recurrent HNSCC. In contrast to T cell deficiency, the recurrent tumors were significantly enriched for markers of tumor-associated neutrophils (TANs; eg., CD177, ELANE), tumor-associated macrophages (TAMs; eg., MRC1, CD68), and immune suppressive myeloid cells (MDSCs; eg., ARG1, CXCR4). Although HPV-specific B cell responses have been associated with HPV[+] HNSCC[36], no significant change was detected for the B cell Fges between the recurrence and non-recurrence groups in our analysis beyond an overall reduction in B cell markers in the tumor compared to the tonsil control. In contrast, a moderate decrease in MHC-I and MHC-II markers was observed in the recurrent group, suggesting that deficient antigen presentation is likely associated with recurrence. Collectively, these data indicate that the recurrent tumors possess an inflammatory and immune suppressive TME characterized with functional depletion of CD8[+] T cells and a marked increase in TANs, TAMs, and MDSCs.

In addition to the immune cells, the recurrent and non-recurrent tumors showed remarkable differences in the tumor stroma, including cancer-associated fibroblasts (CAFs) and ECM. A significant increase was observed for CAF markers, including PDGFRB, FAP and collagens; ECM markers, including laminins, VTN (vitronectin), FN1 (fibronectin 1); and ECM remodeling metalloproteinases, including MMP1, MMP2, MMP3, MMP7, and MMP9 (Fig. 2a). This suggests that the recurrent tumors are characterized with a rich and highly dynamic ECM, and thereby likely being more desmoplastic than the non-recurrent tumors. Moreover, we found a significant enrichment in the endothelium markers MMRN1 and MMRN2, suggesting enhanced angiogenesis potential. This, together with a pro-fibrotic and dynamic ECM, would contribute to increased metastasis for the recurrent tumors. Intriguingly, no significant difference was observed for the tumor proliferation markers MKi67, MCM2, and MCM6, suggesting that recurrence is driven primarily by TME

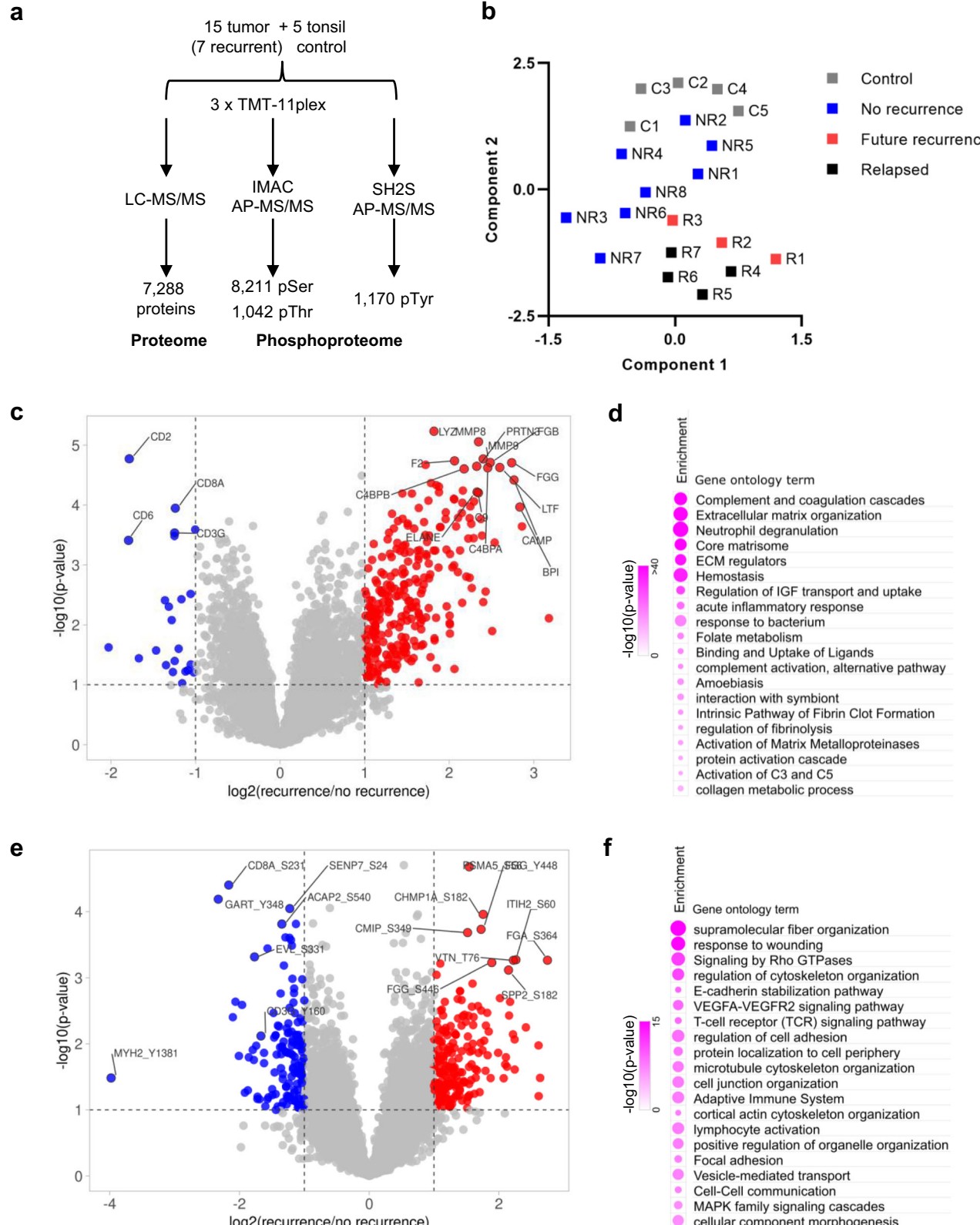

remodeling and the ensuing changes in the crosstalk between the tumor and TME, rather than by an increased proliferation potential of the tumor cells.

Epithelial-mesenchymal transition (EMT) is an important process underlying cancer metastasis, which may play a role in recurrence[5]. To explore this link, we compared the protein levels of EMT regulators that were shown to mediate a partial EMT

(pEMT) program associated with HNSCC lymph nodal metastasis in a recent single-cell transcriptomic study[28]. We found that the majority of the pEMT regulators were significantly increased in the recurrence group. These included the integrin α or β chain proteins ITGA5, ITGA6, ITGB1 and ITGB6, the TGF-β induced protein TGFBI, the serine protease inhibitor SERPINE1, the laminins LAMA3, LAMB3, and LAMC2, and the metalloproteinases MMP1,

**Fig. 1 The proteome and phosphoproteome are distinct between non-recurrent and recurrent HPV$^+$ HNSCC. a** A schematic summary of quantitative mass spectrometry (MS) analysis of HPV$^+$ HNSCC tumor and control tonsil samples. The number of proteins or phosphosites quantified in at least three samples in both the patient and control groups are shown. IMAC immobilized metal affinity chromatography, AP affinity purification, SH2S SH2 Superbinder. **b** The UMAP dimension reduction analysis of the proteome data to show segregation of recurrent (R) from the non-recurrent (NR) tumor proteome. **c** Volcano plot highlighting the significantly increased (red) or decreased (blue) proteins associated with recurrence. **d** Enriched biological processes in the recurrent compared to the non-recurrent tumors based on GO analysis of the corresponding proteome data. The circle size corresponds to the number of genes that belong to the GO term. **e** Significantly increased (red) or decreased (blue) phosphosites associated with recurrence. For panels **c** and **e**, $n = 8$ for the NR group, and $n = 7$ for the R group. **f** Enriched biological processes in the recurrent compared to the non-recurrent tumors based on the corresponding phosphoproteome data.

MMP2 and MMP10, and thrombospondin 1 (THSP1) (Fig. 2b, c). In addition to the elevated pEMT program, a number of markers for hypoxia or stress response were found to be significantly different between the recurrent and non-recurrent groups, implying these processes in recurrence[5,28] (Supplementary Fig. 1).

**Rewiring of the cellular signaling network in the recurrent tumors.** The significant enrichment in ECM and CAF functional markers suggests that the recurrent tumors are characterized with extracellular matrix deposition, remodeling, and crosslinking which together would drive fibrosis to stiffen the stroma and promote malignancy[37,38]. We investigated this further by examining the ECM-integrin/receptor interaction network. Compared to the non-recurrent group, the recurrent group displayed a global and significant increase in essentially all ECM components, including collagen, laminin, fibronectin, tenascin, vitronectin, and thrombospondin (THBS). A concomitant and significant increase in the corresponding receptors, including integrins and CD36, was also observed. Besides increased protein expression, the ECM-receptor network proteins were more highly phosphorylated in the recurrent tumors (Supplementary Fig. 2a, Supplementary Table 7). Together, these data suggest a remarkably extensive and active ECM-receptor interaction and signaling network in the recurrent tumor to promote cell-cell and ECM-tumor interactions, ECM remodeling, and fibrosis.

Because fibrosis plays a critical role in metastasis[38–40] which frequently accompanies recurrence, we compared the key regulators of fibrosis between the NR and R groups. We found that the collagens COL5A3 and COL12A1 and fibronectins 1 (FN1) were significantly increased in the R samples (Fig. 3). LOXL2, which promotes crosslinking of ECM proteins[38], was also elevated, reinforcing our earlier observation that increased fibrosis potential and stiffness of the stroma are associated with recurrence. Nevertheless, MMP8, MMP9 and TIMP1, a metalloproteinase inhibitor[41], were found simultaneously increased with recurrence, suggesting that activity of the MMPs was dynamically regulated. Fibrosis is often a consequence of chronic inflammation[38,39]. In agreement with this assertion, we found the inflammatory cytokine IL1-β significantly increased in the recurrent group.

Complement and coagulation were among the most significantly upregulated process associated with the recurrent tumors from our DEP analysis. Because the coagulation cascade is intimately associated with inflammation and fibrosis[42], we examined the complement system and coagulation cascade and found that numerous components of this cascade were indeed increased in the recurrent tumors (Fig. 4a). Notably, the blood coagulation initiation factors F3 and F12, the coagulation factor F10, prothrombin (F2) and the fibrinogen subunits FGA, FGB and FGG were all significantly increased in the recurrent tumors, suggesting that both the intrinsic and extrinsic coagulation pathways are activated[43]. Similarly, numerous proteins of the complement system, including C2, C3, C5 and CFI (FI), were increased in the R group (Fig. 4b), implicating inflammatory

response mediated by the complement system in HPV$^+$ HNSCC recurrence[44]. These data suggest ECM remodeling, activation of the coagulation, and fibrin formation/fibrinolysis play an important role in remodeling the TME to facilitate HNSCC recurrence.

In contrast to the upregulation of the ECM and coagulation and complement signaling network, the T cell receptor (TCR) signaling pathway was significantly suppressed in the recurrent tumors. Specifically, numerous proteins involved in TCR proximal signaling, including CD3γ, CD3ζ, LCK, and CD45, the key phosphatase for LCK activation, and CD48, a receptor involved in T cell activation, were all significantly decreased in expression and/or phosphorylation (Supplementary Fig. 2b, Supplementary Data 1). Similarly, a number of TCR downstream signaling regulators or effectors, including VAV1, PKCθ, NFAT, P38, and Erk, were down-regulated. These observations, which suggest defective TCR signaling, reinforce our previous observation that the recurrent tumors were defective in T cell function. Besides the TCR signaling pathway, numerous proteins of the spliceosome were seen down-regulated in the recurrent tumor, implicating defective RNA splicing in recurrence (Supplementary Fig. 3). In this regard, alternative splicing has been shown to play an important role in tumor progression, invasion, metastasis, and angiogenesis[45].

**Kinome reprogramming associated with recurrence.** Protein kinases play a critical role in tumorigenesis and cancer metastasis, which has provided a basis for kinase-targeted therapy in the treatment of numerous cancers. In fact, >90% of HNSCC is characterized with overexpression of the EGF receptor (EGFR) tyrosine kinase (TK)[5,46]. To find out if EGFR or other TKs and STKs (Ser/Thr kinases) are involved in HNSCC recurrence, we compared the kinase expression and phosphorylation levels between the recurrent and non-recurrent groups. Indeed, numerous kinases exhibited significant differences between the R and NR groups (Fig. 5a, b). Specifically, the receptor tyrosine kinases (RTKs) EGFR, PDGFRB, MST1R, and PTK7 were found significantly increased in expression in the recurrent tumors (Fig. 5a, b). PDGFRB plays a critical role in CAF proliferation while DDR1 is a collagen receptor[47]. Together these two RTKs may play an important role in regulating fibrosis[47,48]. The macrophage-stimulating protein receptor MST1R (or RON), which regulates the migration and activation of macrophages and activates wound-healing responses[49], may play the dual role of promoting TAM and MDSC recruitment and fibrosis. The protein tyrosine kinase 7 (PTK7) has been recently shown to mediate the crosstalk between the tumor cells and CAFs and to promote cancer stemness in head and neck cancer[50]. Its upregulation in recurrent tumors suggests that increased cancer stemness is associated with recurrence. In addition to the RTKs, the cytoplasmic tyrosine kinases (CTKs) YES1, which may be recruited to activated EGFR and PDGFRB, was also significantly upregulated in the R group. YES1 activation may be corroborated by decreased CSK, a negative regulator of the Src family kinases.

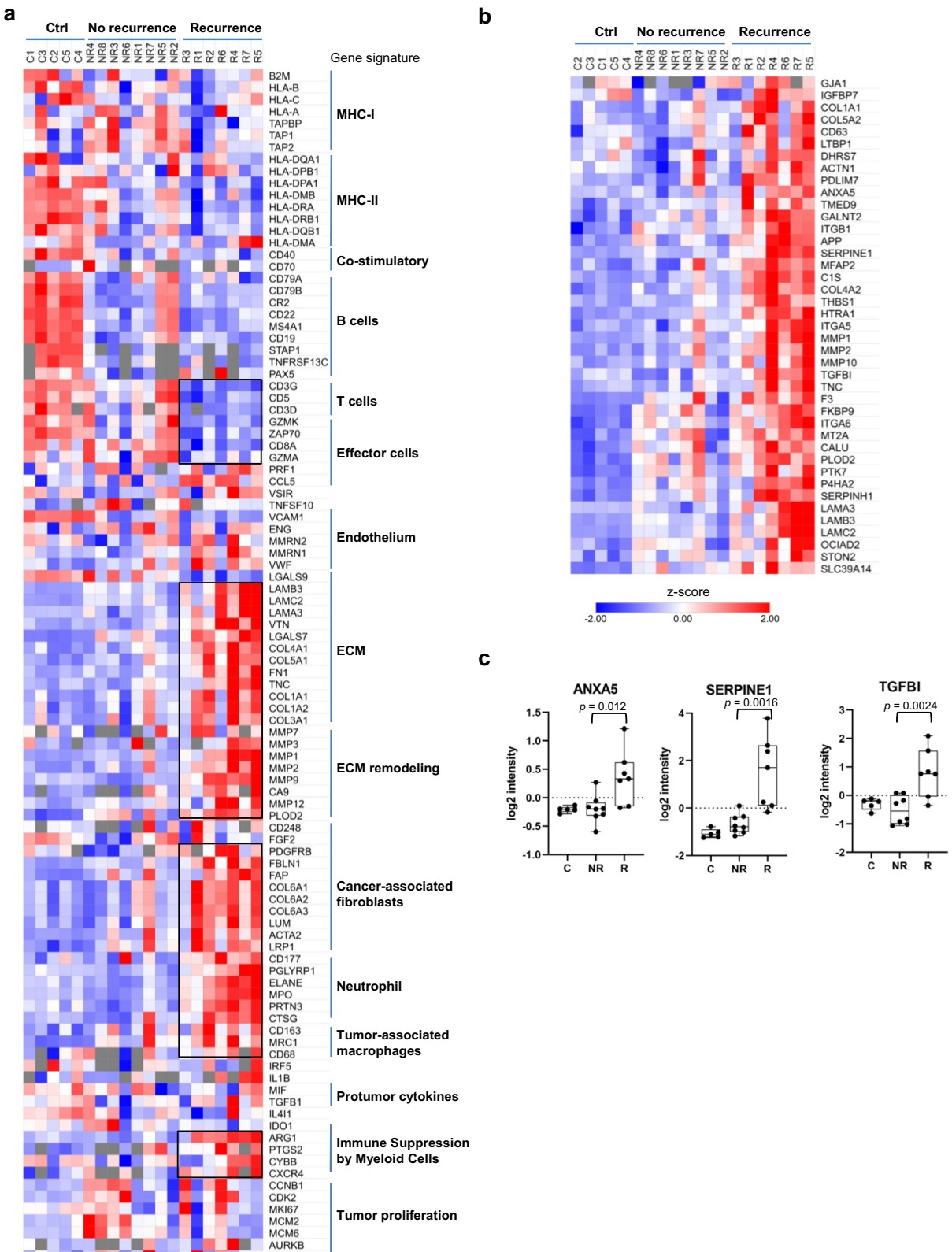

**Fig. 2 Remodeling of the tumor microenvironment (TME) underpins recurrence. a** Heatmap of functional gene signatures (Fges) to show the significant changes in cellular or extracellular components (boxed) in the recurrent compared to the non-recurrent groups and tonsil controls. ECM extracellular matrix. **b** An increased EMT potential is associated with recurrence. The recurrent tumors were characterized with significantly increased expression of the markers for a partial EMT program (*p* < 0.1 between the NR and R groups). **c** Representative examples of pEMT markers with significant differences in expression between recurrent and non-recurrent samples. The box in a the box plot extends from the 25th to the 75th percentile. The whiskers go down to the smallest value and up to the largest. For the panels **b** and **c**, *n* = 8 for the NR group, and *n* = 7 for the R group.

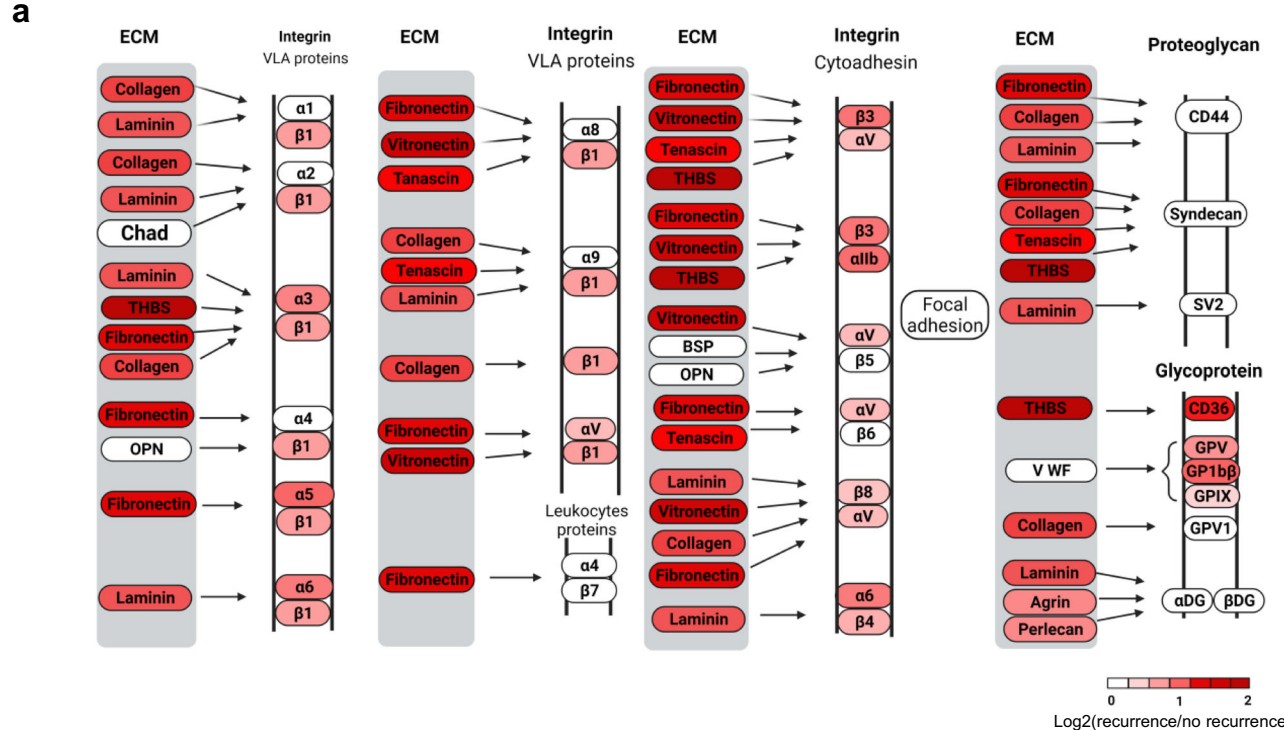

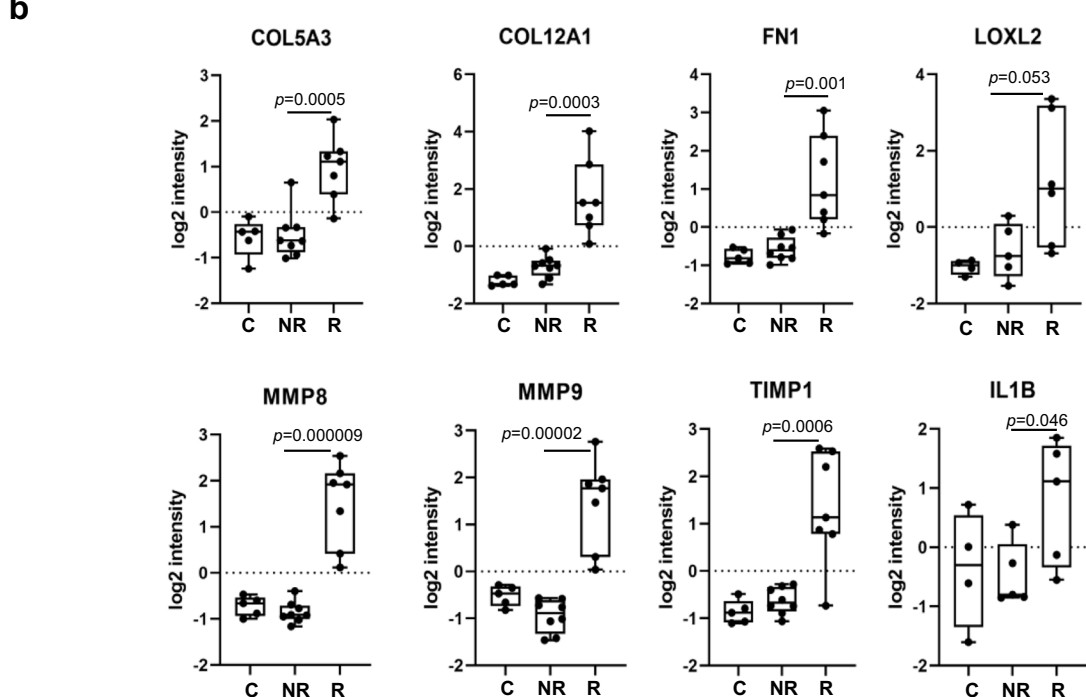

**Fig. 3 Elevated extracellular matrix (ECM) signaling is associated with recurrence. a** The ECM ligand-receptor interaction network is up-regulated in the recurrent tumors. The diagram was recreated based on the KEGG pathway ID hsa04512. Changes in the protein level (Log2(R/NR)) is color-coded with red indicating increased protein expression. VLA very late antigen. **b** Representative examples of ECM and/or fibrosis regulators with significant differences in protein expression between the R and NR groups. *P* values shown were based on Student's *t*-test. The box in the box plot extends from the 25th to the 75th percentile. The whiskers go down to the smallest value and up to the largest. *n* = 8 for the NR group, and *n* = 7 for the R group.

Elevated phosphorylation of DDR1, PTK2 (FAK), EGFR, and FER was also detected in the R group, suggesting increased integrin, collagen, and cytoskeletal signaling in the recurrent tumors. Notwithstanding these observations, the T/B cell protein kinases LCK, LYN, BTK, and SYK, and the cytokine signaling

kinases JAK1, TYK2, and JAK3 all had decreased phosphorylation in the R group, consistent with compromised T cell and B cell function in the recurrent tumors.

In addition to the TKs, a large number of STKs were significantly differentially expressed in the recurrent tumors.

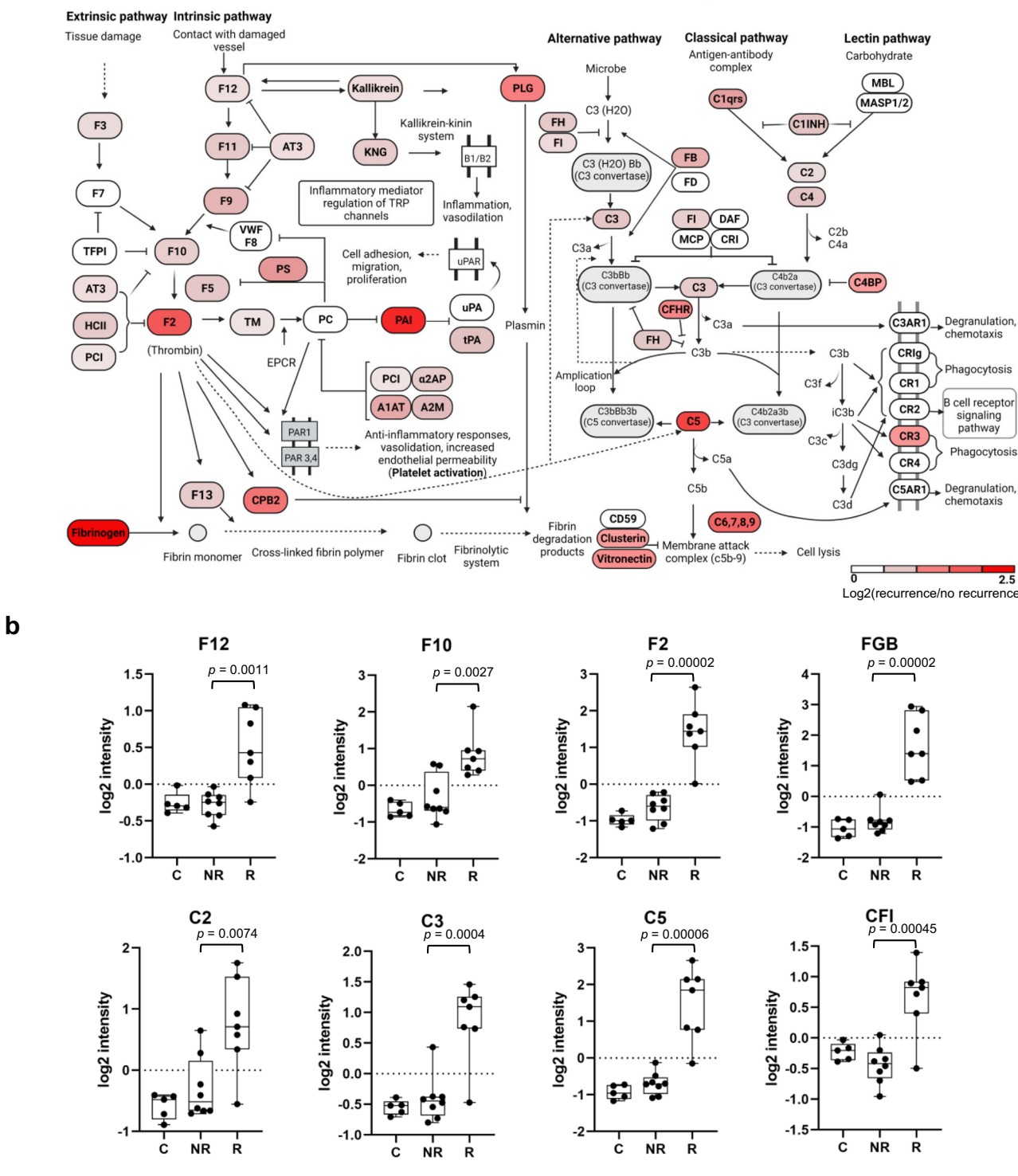

**Fig. 4 The coagulation process and complement system are up-regulated in recurrent HNSCC. a** The complement and coagulation cascades showing overexpression (red marked) of key pathway components in the recurrent samples. The diagram was recreated based on the KEGG pathway ID hsa04610. **b** Examples of coagulation or complement regulators that are significantly different between the R and NR groups. P values were based on Student's t-test. The box in the box plot extends from the 25th to the 75th percentile. The whiskers go down to the smallest value and up to the largest. $n = 8$ for the NR group, and $n = 7$ for the R group.

The STKs with increased protein levels included DAPK3, CDC42BPB, TRIO, ROCK2, which are regulators of cytoskeleton reorganization and cell migration, and CSNK1D (CK1D), CSNK2A2 (CK2A2) and AKT2, which are regulators of cell survival[51]. Moreover, numerous STKs exhibited elevated (Fig. 5c) or reduced (Supplementary Fig. 4) phosphorylation in the recurrent tumors. The over-phosphorylated STKs included those that regulate cell proliferation and survival (eg., MAP2K2, MAPKAPK5, MAPK9, PRKD1, BRAF, AKT2, TTK, and WNK1), cytoskeletal reorganization/cell migration (eg., MYLK,

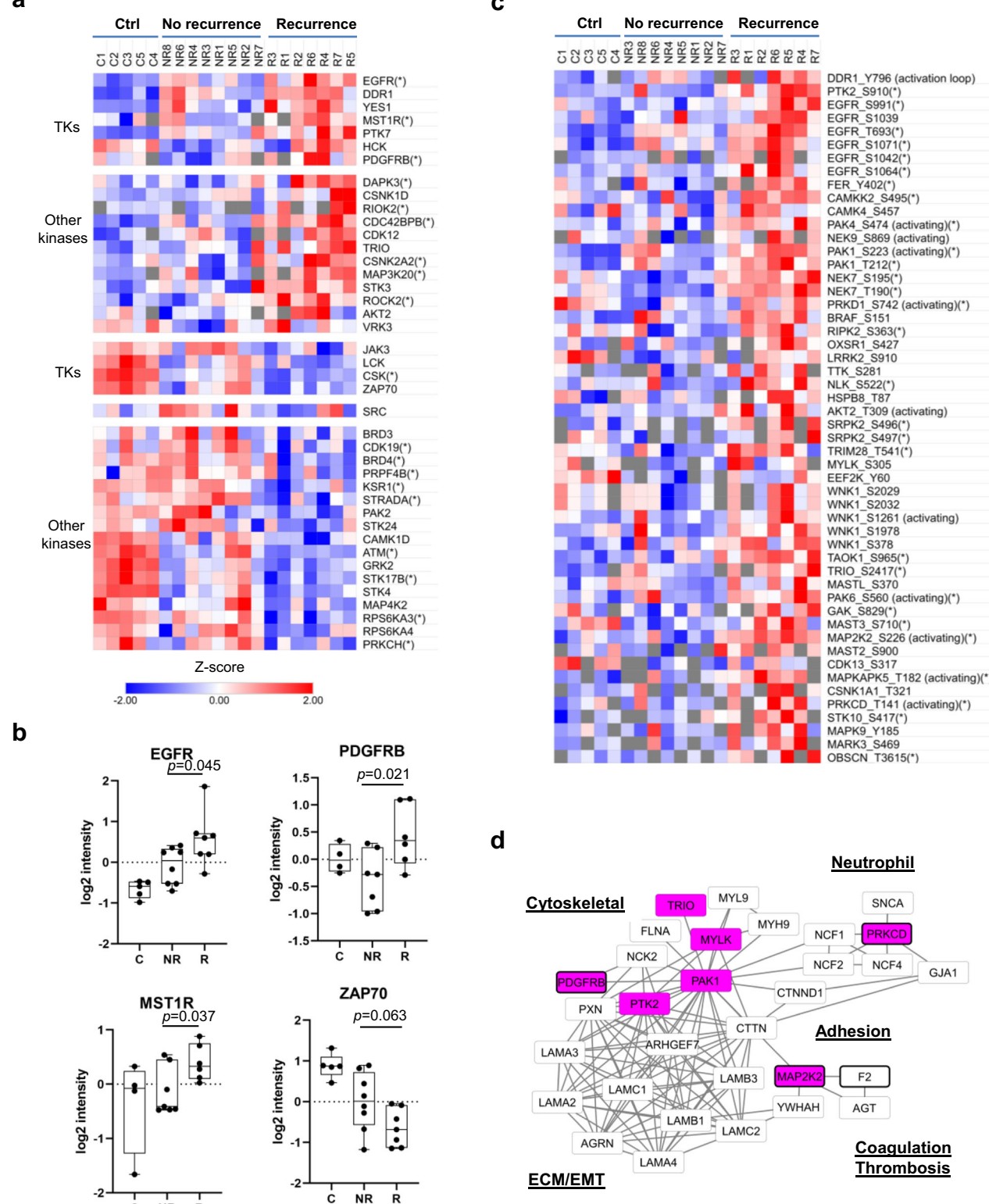

TRIO, PAK1, PAK4, PAK6, OXSR1, MARK3, NEK7) or Ca$^{2+}$-dependent signaling (eg., PRKCD, CAMKK2, CAMK4 and LRRK2).

Notwithstanding these observations, we found a significant decrease in the pro-apoptotic kinases STK24, STK4, STK17B, RPS6KA3/4 and ATM, the tumor suppressor STRADA, and PRPF4B (PRP4), a critical regulator of RNA splicing[52]. The phosphorylation levels of these proteins were also significantly reduced in the recurrent tumors (Supplementary Fig. 4). Furthermore, a significant decrease in phosphorylation was observed for the TCR and BCR (B cell receptor) proximal kinases LCK and SYK at their respective activation loop Tyr sites, indicating, once again, decreased T/B cell function in the recurrent tumors.

Collectively, these data suggest that a broad reprogramming of the kinome at both the protein and phosphorylation level

**Fig. 5 Kinome reprogramming in the recurrent tumor. a** Heatmap of protein kinases with significantly increased or decreased expression in the recurrent tumors (>2-fold difference and $p < 0.1$ between the NR and R groups). An asterisk (*) next to the kinase name indicates $p < 0.05$. TK tyrosine kinase. **b** Representative examples of kinases (TK or STK) with significant differences in expression between the NR and R groups. The box in the box plot extends from the 25th to the 75th percentile. The whiskers go down to the smallest value and up to the largest. **c** Heatmap of protein kinases with significantly increased phosphorylation in the recurrent tumors compared to the non-recurrent counterparts (>2-fold increase in the R group and $p < 0.1$ between the NR and R groups). An asterisk (*) next to the phosphosite name indicates $p < 0.05$. **d** A reprogrammed TK/STK signaling network nucleated by PAK1 and FAK regulates multiple cellular and biological processes associated with recurrence. The nodes shown consist of proteins or phosphosites that are increased in the recurrence samples (>1.5-fold increase and $p < 0.1$ between R and NR groups). In all the panels, $n = 8$ for the NR group, and $n = 7$ for the R group. ECM extracellular matrix, EMT epithelial-mesenchymal transition.

underlies recurrence. Importantly, the reprogrammed TKs/STKs form an extensive protein-protein interaction and signaling network that regulate the multiple processes that are altered in the recurrent tumors, including cytoskeletal reorganization, cell adhesion, ECM remodeling, EMT, neutrophil function and coagulation/thrombosis (Fig. 5d).

**Predictive biomarkers and potential therapeutic targets for recurrent HNSCC.** Besides generating mechanistic insights, a major motivation of this study is to identify predictive biomarkers for patient stratification and novel therapeutic targets. Of the hundreds of differentially expressed and/or differentially phosphorylated proteins, many are potential therapeutic targets based on the DrugBank, a database of FDA-approved therapeutic targets[53]. The list of therapeutic targets identified herein contained not only protein kinases such as PDGFR, EGFR, MAP2K2, PDKCD, and JUN, but also regulators of fibrosis, coagulation/complement systems, neutrophil function, and inflammation, including MMP9, fibrinogens, PROS1, C5, C3, ELANE, FCGR3A and IL1B (Fig. 6a, b). MMP9, which tops the list of candidate therapeutic targets, plays an essential role in local ECM degradation, leukocyte migration and EMT. Because aberrant regulation of these processes underlies recurrence of HPV$^+$ HNSCC, targeting MMP9 and its regulators may be a plausible approach by which to curb recurrence[54]. In this regard, the expression of MMP9 is found to be controlled by an elaborate PPI network that include the upstream regulators EGFR, PTK2, MAPK9, IGFP7, IL1B, STAT5A, FOS/JUN, and CCL5 (Fig. 6c). While EGFR, PTK2 (FAK) and MAPK9 may directly regulate FOS, the cytokine IL1B[54] may regulate JUN directly or indirectly via CCL5 which also plays a critical role in monocyte recruitment and TAM infiltration within tumors[55,56]. Therefore, targeting these regulators, especially when in combination, provides an alternative means to targeting MMP9.

Personalized treatment of HNSCC based on recurrence potential would greatly enhance the efficacy of treatment for the recurrent cases and reduce morbidity of patients with non-recurrent cancer. Towards this goal, we next aimed at identifying predictive biomarkers of recurrence, by investigating a correlation between our proteomic observations and clinical samples from other cohorts. Because a database for HNSCC recurrence is unavailable, we resorted to predict patient survival using the TCGA HPV$^+$ HNSCC database[57] ($n = 71$). A predictive marker was chosen when the following three conditions were met: (1) the protein abundance was significantly different between the NR and R groups, (2) the protein is an FDA-approved drug target (Fig. 6a, b), and (3) the expression of the marker is significantly correlated with patient survival. Our analysis identified eight potential biomarkers of HNSCC recurrence, namely PROS1, ANXA3, COL2A1, F3, TUBB3, PSMB5, SLC7A11/xCT, and CD2 (Fig. 7a). Most of these proteins are involved in the identified fibrosis or coagulation pathways/PPI networks associated with recurrence (Figs. 3, 4). The SLC7A11/xCT pSer26 site is a substrate of the kinase complex mTORC2[58], whose activation has been implicated in HNSCC

disease progression[59]. Kaplan-Meier analysis showed that reduced expression of any of these markers, except for CD2, was significantly associated with improved disease-specific survival. In contrast, a high expression level of the T cell marker CD2 was correlated with better outcome (Fig. 7b).

## Discussion

Our comparative proteomic and phosphoproteomics analysis of recurrent and non-recurrent tumors indicate the tumor microenvironment (TME) plays an essential role in HPV$^+$ HNSCC recurrence. The TME comprises immune cells, blood vessels, and the stroma, including cancer-associated fibroblasts (CAFs) and the extracellular matrix[60]. We found that the recurrent tumors differ significantly from the non-recurrent tumors in these hallmark features. First, the recurrent tumors are characterized with an inflammatory and suppressive tumor immune microenvironment (TIME)[61], characterized with increased TANs and TAMs, decreased antigen presentation, and deficient T/B cell function[62]. This suggests that the immune evasion by the recurrent tumor is mediated by the lymphoid and myeloid cells[63]. The enrichment of TANs is consistent with a recent observation that later stage oral squamous cell carcinoma patients have elevated plasma neutrophil extracellular traps (NETs)[64]. This also provides an explanation for hypercoagulability, a major cancer-associated complication linked to poor patient prognosis, observed in the recurrent samples. Second, we found a remarkable and significant increase in CAFs in the recurrent group, suggesting fibrosis or a proclivity for fibrosis underlies recurrence. The CAFs are not only major contributors of cancer fibrosis through increased collagen crosslinking, but they may also promote tumorigenesis through the collagen-integrin and collagen-RTK signaling networks. In the same vein, the role of CAFs in promoting EMT has been observed in multiple cancer types[65]. Third, the TME is enriched in ECM and ECM remodeling proteins. ECM deposition, remodeling and crosslinking can drive fibrosis to stiffen the stroma, which, in turn, enhance tumor progression, aggression, angiogenesis, and hypoxia[37,66]. Matrix-rich, fibrotic tumors are also associated with poor immunity characterized with poor T cell penetration and activation. Finally, the crosstalk between the tumor cells, CAFs, immune cells and ECM may promote tumor progression, tumor cell intravasation and metastasis, leading to cancer relapse[37,38,67]. Indeed, we found that the recurrent tumors were enriched in genes mediating a partial epithelial-to-mesenchymal (pEMT) program, reminiscent of the finding from a recent single cell RNA analysis of matched primary and lymph node metastasis HNSCC (mainly the HPV$^-$ subtype)[28]. This suggests that an increased EMT potential is associated with HNSCC recurrence in general regardless of the HPV infection status.

TME remodeling is likely driven by the corresponding changes in the underlying signaling pathways and/or protein-protein interaction networks. That essentially all components of the ECM-integrin signaling network are significantly over-expressed and hyper-phosphorylated suggest ECM dynamics and remodeling are cardinal features of HNSCC recurrence. The involvement of the coagulation/thrombosis pathway and the

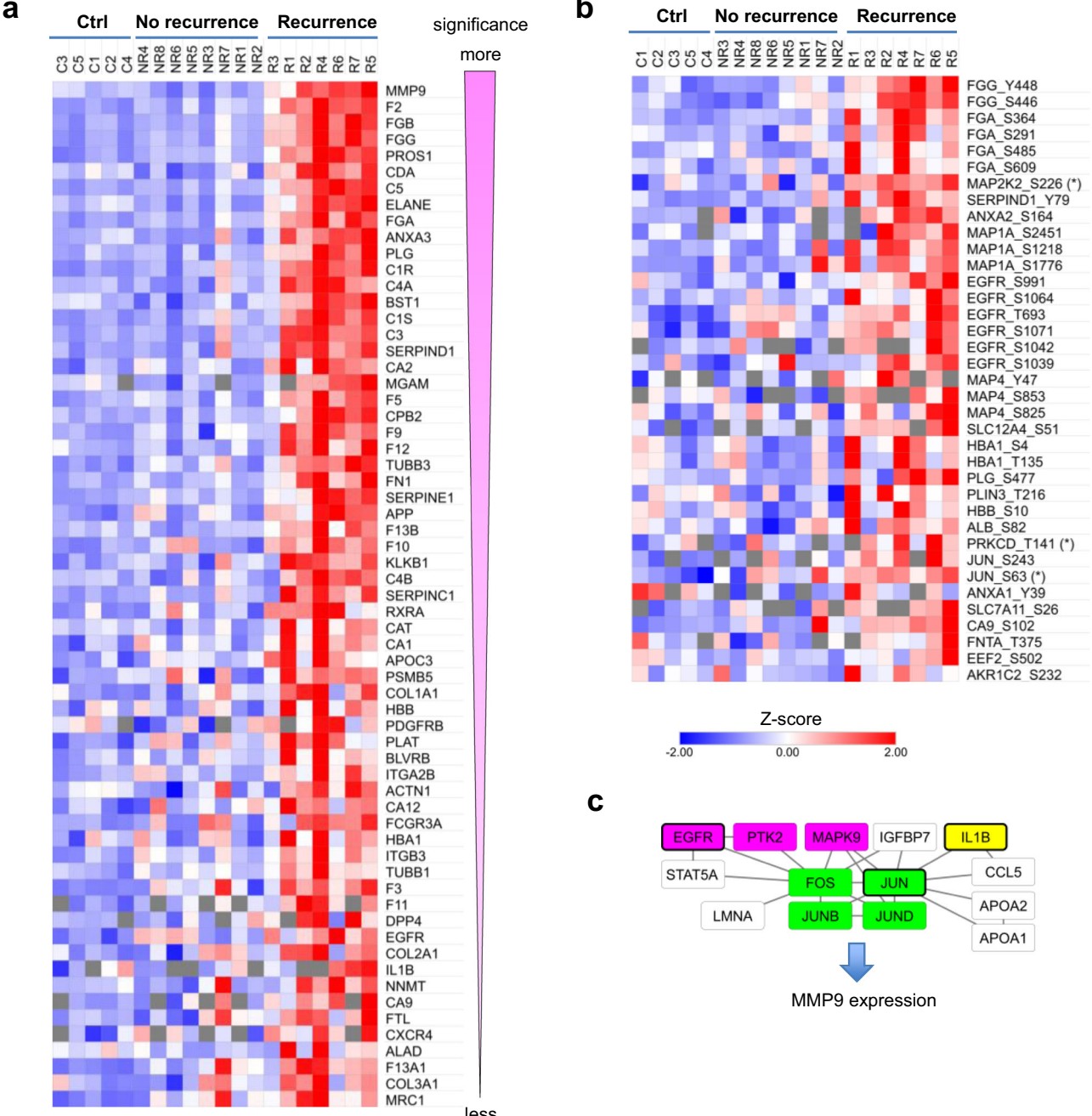

**Fig. 6 Candidate therapeutic targets for precision treatment of recurrent HPV⁺ HNSCC. a** Heatmap showing overexpression of a list of FDA-approved therapeutic targets in the recurrent tumors. The proteins are sorted, from top to bottom, with the corresponding p values calculated from the proteome data between the NR and R group. Proteins with >1.5-fold increase in the R group and *p* < 0.1 between R and NR groups are included. EGFR and PDGFR are among the significantly upregulated therapeutic targets. **b** A list of FDA-approved drug targets with a significant increase in phosphorylation in the recurrent tumors. Asterisk(*): kinase activity-inducing residue (e.g., MAP2K2, PRKCD, c-Jun phosphosites) based on the PhosphositePlus database. Phosphosites with >1.5-fold increase in the R group and *p* < 0.1 between R and NR groups are included. **c** A kinase and cytokine signaling network that regulates increased MMP9 expression in the recurrent tumors. The nodes consist of proteins or phosphosites increased in the R group (>1.5-fold increase and *p* < 0.1 between the R and NR groups). Kinases are shown in magenta. Components of the AP-1 transcription factor complex, c-Jun/JunB/JunD/c-Fos, are in green. In all the panels, *n* = 8 for the NR group, and *n* = 7 for the R group.

complement system in recurrence may be related to HPV infection. In this regard, it would be of interest to confirm this assertion by proteomic analysis of an HPV⁻ HNSCC cohort. Depletion of T cells has been associated with recurrent HNSCC, and our study suggests that both tumor infiltration and function of CD8⁺ T cells are compromised in the HPV⁺ HNSCC. A recent study has identified the spliceosome pathway as a key cellular

function elevated in esophageal squamous cell carcinoma[68]. However, our proteomic data suggests that the RNA splicing pathway is compromised in the recurrent HNSCC.

Because EGFR is overexpressed in 80–90% HNSCC, we investigated the role of protein kinases and kinase signaling in recurrence. Indeed, we have identified significant changes in the expression and/or phosphorylation of numerous TKs and STKs,

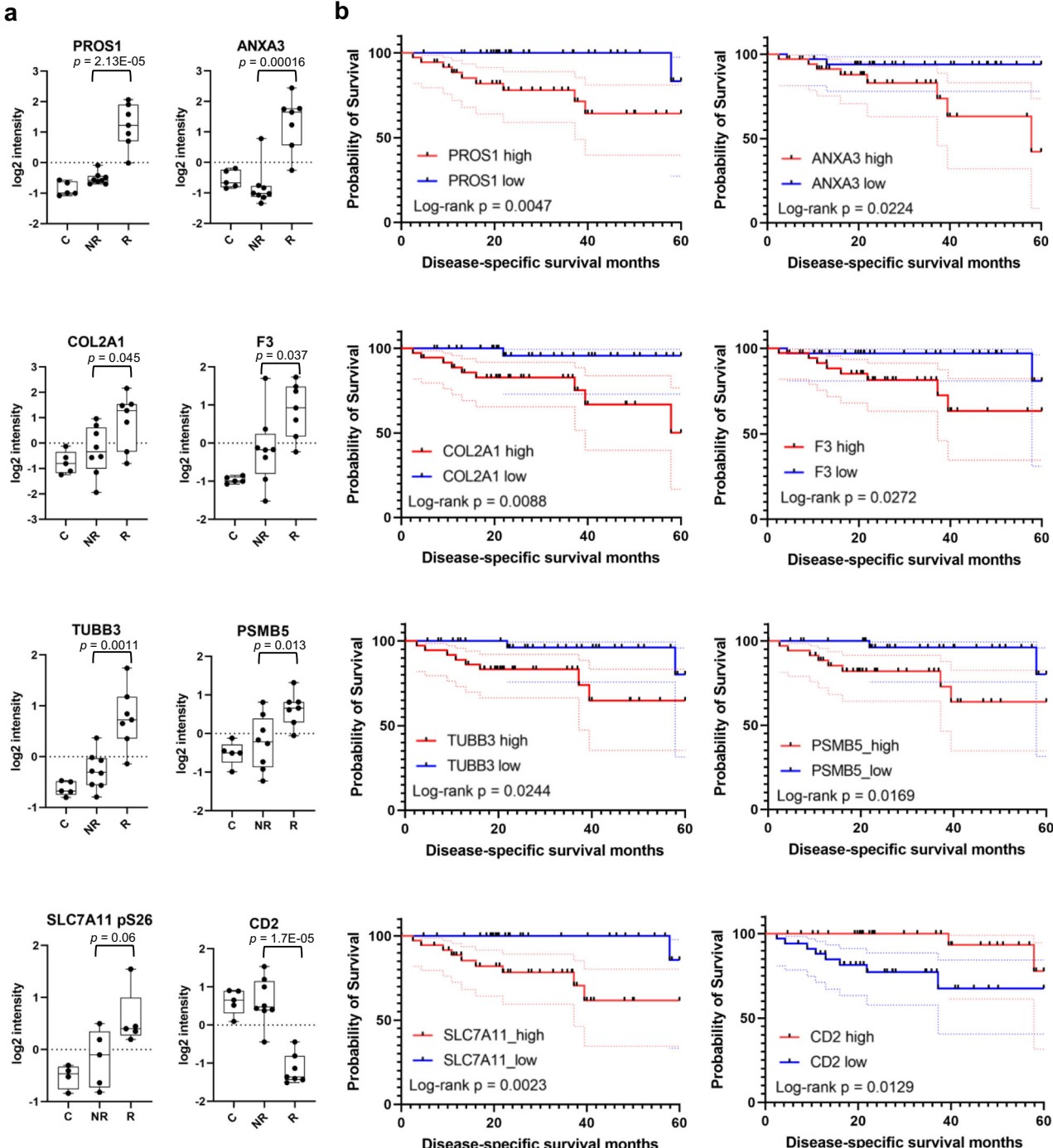

**Fig. 7 Potential biomarkers for predicting recurrence and survival. a** Box plots of representative biomarkers emerging from this study. *P* values shown were based on Student's *t*-test. FDA-approved drugs are available for the upregulated proteins shown here. The box in the box plot extends from the 25th to the 75th percentile. The whiskers go down to the smallest value and up to the largest. *n* = 8 for the NR group, and *n* = 7 for the R group. **b** Kaplan-Meier analysis showing that low expression of the fibrosis and inflammation markers or high expression of the T cell marker CD2 are correlated with longer survival. P values were based log-rank test. The TCGA HPV⁺ HNSCC (*n* = 71) cohort was used in the survival prediction. For each gene, the 71 patients were divided into high expression (*n* = 36) or low expression (*n* = 35) group, and the disease-specific survival months were plotted. The dotted lines show 95% confidence intervals.

suggesting extensive kinome reprogramming associated with recurrence. Of note, an altered kinase signaling network involving EGFR, PDGFR, PAK1, PTK2 (FAK), PRKCD, and MAP2K2 appear to regulate the aberrant changes in the cancer cells and the TME accompanying recurrence.

Our investigation into the mechanism of HNSCC recurrence led to the discovery of numerous potential therapeutic targets.

Some targets, including EGFR, PDGFR, FAK, CXCR4, and MMP9, have corresponding inhibitors that have been approved or are under active clinical trials for cancer treatment. Our study suggests that these inhibitors may be repurposed for the treatment of recurrent HPV⁺ HNSCC. Although the EGFR antibody cetuximab is generally used in combination with radiation in HPV⁻ HNSCC where cytotoxic chemotherapy is not the best

option due to comorbidities, our data suggests that cetuximab in combination with radiotherapy or chemotherapy may be used to preempt or treat recurrent HPV[+] HNSCC. In the same vein, FDA-approved PDGFR inhibitors may be used in the above combination therapies owing to its critical role in promoting tumorigenesis and fibrosis. Because both EGFR and PDGFR signal through MAP2K2 (MEK2), the latter of which is significantly activated in the recurrent tumors, pharmacological blockade of MAP2K2 activity may be an alternative approach for the treatment of recurrent HPV[+] HNSCC. Furthermore, our work led to the identification of >30 potential FDA-approved therapeutic targets for exploration and validation[53].

Checkpoint blockade immunotherapies benefit only 20–30% of HNSCC patients[61]. This moderate rate of response demands for better understanding of the immune landscape of HNSCC. The recurrent tumors may be more resistant to immune checkpoint inhibitors (ICI) as collagen can promote anti-PD-1/PD-L1 resistance via CD8[+] T cell exhaustion[69]. Reducing collagen deposition by inhibiting LOXL2, which promotes crosslinking of ECM proteins and is significantly upregulated in the recurrent HNSCC, maybe a potential approach to activate the exhausted CD8[+] T cells in combination with PD-1/PD-L1 blockade immunotherapy as demonstrated recently for lung cancer in a mouse model[69]. In this regard, it is worth noting that an LOXL2 blocking antibody is currently in preclinical and clinical fibrosis trials[66]. Other CAF-targeting drugs that are currently under clinical trials may also be repurposed for the treatment of recurrent HNSCC, including inhibitors of FAK (PTK2), ROCK, and CXCR4 which we have found to be significantly overexpressed and/or hyper-activated in the recurrent tumors. While FAK and ROCK are key regulators of the cytoskeleton, CXCR4 is expressed by monocytes or monocytic MDSCs[55]. Therefore, the TME provides a wealth of potential therapeutic targets for future exploration. Our work further suggests that, apart from targeting exhausted T cells, the myeloid cell population affords an attractive immunotherapeutic target for the treatment of recurrent HPV[+] HNSCC.

Our recurrent tumor samples contained both primary tumors with future recurrence and resected tissues from relapsed tumors. That the proteome and phosphoproteome of these two sets of samples converge on the key features identified herein suggest that it is possible to predict recurrence using the primary tumor biopsy prior to disease relapse. In support of this, several proteins identified to play an important role in recurrence are capable of predicting patient survival using an independent TCGA HNSCC cohort. These potential biomarkers, including PROS1, ANX3, F3, COL2A1, TUBB3, PSMB5, SLC7A11/xCT and CD2, are key regulators of fibrosis, coagulation, metastasis, or immune responses. PROS1 and F3 are particularly attractive biomarkers because they are secreted proteins. Nevertheless, it remains to be determined if these proteins are also elevated in the peripheral blood of patients with recurrent disease.

A key limitation of our study is the modest sample size. Despite this, our proteomic and phosphoproteomic analysis has identified a number of TME features associated with recurrence. Our study highlights the need to conduct deep and quantitative proteomic analysis independently or in combination with transcriptomic profiling to better understand the systems basis of cancer recurrence and/or metastasis to inform prognosis and treatment of recurrent/metastatic cancers.

## Data availability

Source data for the figures are provided as Supplementary Data 2, 3. The mass spectrometry proteomics/phosphoproteomics data have been deposited to the ProteomeXchange Consortium via the PRIDE partner repository with the dataset identifier PXD030343. All other data are available upon request.

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

## Acknowledgements

This study was funded by Canadian Institutes for Health Research (CIHR) grants MOP 201809 to S.S.C.L. and MOP 340674 to A.C.N. and a Canadian Cancer Society Discovery Grant to S.S.C.L. S.S.C.L. held the Canada Research Chair (Tier I) and Wolfe Medical Research Professorship in the Molecular and Epigenetic Basis of Cancer. A.C.N. held the Wolfe Surgical Research Professorship in the Biology of Head and Neck Cancers. R.A. was supported by a Scholarship from the Canadian Cancer Society and X.L. was supported by a Mitacs Post-Doctoral Fellowship. We thank Dr. Mingliang Ye (Dalian Institute of Chemical Physics) for providing the IMAC resin used in this study and Dr. Sally Ezra (Western University) for T cell preparation.

## Author contributions

S.S.C.L. and A.C.N. conceived the study. S.S.C.L. and T.K. designed the experiments. T.K., X.L., P.Y.F.Z., and J.W.B. performed the experiments. T.K., S.S.C.L., P.Y.F.Z., R.A., and Q.Z. analyzed the data. S.S.C.L. and T.K. wrote the manuscript with inputs from P.Y.F.Z. and A.C.N. All authors approved the manuscript prior to submission.

## Competing interests

The authors declare no competing interests.
