## [Peer Review File · Communications Medicine]

Reviewers' comments:

Reviewer #1 (Remarks to the Author):

The authors have performed a proteomic analysis of HPV+ oropharyngeal cancers, recurrent and nonrecurrent, to look for key differences. Many interesting analyses were performed, and as stated, it is more of a hypothesis-generating type of study.

There were several aspects that need clarification:

1) Table S1 is absent from the materials. The defined patient population is lacking important details. Were these patients smokers? What type of treatment did the recurrent patients receive? Did the 4 recurrent patients receive surgery, or are these just biopsy specimens?

2) For the remaining analyses, treating 'recurrent' tumors is different than those that were treated, I believe? Again, it is not clear whether those tumors had been radiated, or received adjuvant chemotherapy. One would expect any degree of treatment itself to skew the results. Fibrosis, alteration in TME, etc. might be explained by local treatment effects.

3) Did any of these patients receive immunotherapy as part of their regimen? Again, this would alter the subsequent analyses as well. Hopefully this is covered in table S1.

4) I appreciate the subsequent analyses, as they cover many of the questions that would be important to answer. However, there was no validation or correlation with clinical samples to understand the treatment or prognostic value of their findings.

5) The authors do recognize that this is a small sample size. These are difficult samples to procure, and the analyses were certainly in depth. That is why the uniformity of the samples needs to be well understood. To answer such a focused question, the samples do need to be well curated to minimize factors that might introduce unintended variability.

Reviewer #2 (Remarks to the Author):

Manuscript ID: COMMSMED-22-0636-T

Title: Proteome and phosphoproteome signatures of recurrence for HPV+ head and neck squamous cell carcinoma

Comments to Authors:

In the current study, the authors employed proteomic and phosphoproteomics approaches to examine 4 non-recurrent, 3 primary tumors that later recurred and 4 recurrent tumors of HPV+ head and neck squamous cell carcinoma (HNSCCs). HPV+ HNSCCs in general have better prognosis; however, a fraction of these patients develop recurrence and have much worse prognosis. Hence, this study addresses a highly important question in the field, namely, what are the differences between recurrent and non-recurrent HPV+ HNSCCs. The authors found that the pathways related

to TME remodeling (ECM, complement etc.) show significant differences. Moreover, recurrent tumors show defects in T cell functions and TCR signaling. The study will provide interesting and new information to the field of HPV+ HNSCCs and help to further elucidate the mechanisms of recurrence and define predictive markers.

Point-by-point comments:

- 1) In Figure 5, the significant proteins were shown with $p < 0.1$ between the NR and R groups. What if $p < 0.05$, any protein expression will be significantly different between NR and R?
- 2) It is unclear why only a few genes (PROS1, ANXA3, COL2A1, F3, TUBB3 and CD2) were chosen for predictive marker studies on page 11.
- 3) Discussion is overly long and often repeated result description, which can be shortened.
- 4) As the authors pointed out, the sample size is limited. However, the results are of great interest and should provide new information for the field to better elucidate the recurrence mechanisms.

Minor issues:

- 1) There are some typos such as page 11, line 13, "Because aberrant regulation of these processed underlie recurrence of HPV+ HNSCC..."

Reviewer #3 (Remarks to the Author):

The manuscript by Kaneko et al., entitled "Proteome and phosphoproteome signatures of recurrence for HPV+ head and neck squamous cell carcinoma" characterized recurrent and non-recurrent HPV+ HNSCC using MS-based proteomic and phosphoproteomic analysis. The authors have found significant differences related tumor microenvironment and epithelial-mesenchymal transition.

Experimental design/method was well described and data analysis was carried out carefully. I have some concerns on the current manuscript that the authors should address.

1. The author employed phosphoproteomic approach to characterize clinical cancer samples. Phosphorylation is sensitive to environmental changes, so it is required to explain the detail of cryopreservation after sample resection and the cryopreservation method.
2. For sample quality check, the author mentioned that "Frozen section analysis was carried out to confirm tumor cellularity greater than 70%." However, NR2 sample is localized in the control sample area in PCA plot Fig1A. From this figure, percentage of cancer cells in the NR2 sample is low. This discrepancy needs to be explained.
3. Boosting technology was used to enhance pY identification. It is very powerful to increase identification number, however, concerned that over boosting by more than a factor of 100 will result in a loss of quantifiability. Therefore, it is recommended to exclude data where the intensity of the boost sample is extremely strong compared to the intensity of the sample.

4. Regarding EMT, were there differences in the expression levels of the typical markers CDH1 and CDH2?

Point-by-point responses to the referees' comments

Reviewers' comments quoted verbatim in italics

Reviewer #1 (Remarks to the Author):

The authors have performed a proteomic analysis of HPV+ oropharyngeal cancers, recurrent and nonrecurrent, to look for key differences. Many interesting analyses were performed, and as stated, it is more of a hypothesis-generating type of study.

There were several aspects that need clarification:

1) Table S1 is absent from the materials. The defined patient population is lacking important details. Were these patients smokers? What type of treatment did the recurrent patients receive? Did the 4 recurrent patients receive surgery, or are these just biopsy specimens?

Response: We have moved all supplementary tables into a single supplementary PDF file. We have included smoking and treatment information in Supplementary Table 1. Four out of the eight non-recurrent patients were smokers, and five out of the seven recurrent patients were smokers. Of the seven recurrent patients, six received cisplatin chemotherapy, and the other one received radiation.

2) For the remaining analyses, treating 'recurrent' tumors is different than those that were treated, I believe? Again, it is not clear whether those tumors had been radiated, or received adjuvant chemotherapy. One would expect any degree of treatment itself to skew the results. Fibrosis, alteration in TME, etc. might be explained by local treatment effects.

Response: All the primary tumour samples (NR1-NR8, and R1-R3) were surgically collected before any treatment. For the four relapsed samples (R4-R7), the patients received cisplatin or radiation treatment until no evidence of disease was observed, but later tumours redeveloped. The R4-R7 samples were collected from the relapsed tumour. Therefore, the patients were not receiving treatments when the recurrence happened. This information was added to the Methods section.

3) Did any of these patients receive immunotherapy as part of their regimen? Again, this would alter the subsequent analyses as well. Hopefully this is covered in table S1.

Response: No patients received immunotherapy. This information was added to the Methods section.

4) I appreciate the subsequent analyses, as they cover many of the questions that would be important to answer. However, there was no validation or correlation with clinical samples to understand the treatment or prognostic value of their findings.

Response: To investigate a correlation between our proteomic observations and clinical samples from other cohorts, we used the TCGA database (n = 71) for Kaplan-Meier survival analysis. Because recurrence information is incomplete in the TCGA database, we used disease-specific

survival as a proxy for recurrence. We indeed observed some correlation between the recurrence markers and survival, as shown in Fig. 7B.

5) The authors do recognize that this is a small sample size. These are difficult samples to procure, and the analyses were certainly in depth. That is why the uniformity of the samples needs to be well understood. To answer such a focused question, the samples do need to be well curated to minimize factors that might introduce unintended variability.

Response: The detailed clinical information was provided in Supplementary Table 1. All patients were infected with HPV16 and the status was confirmed by PCR and Sanger sequencing. Frozen section analysis was carried out to confirm tumor cellularity greater than 70%.

Reviewer #2 (Remarks to the Author):

In the current study, the authors employed proteomic and phosphoproteomics approaches to examine 4 non-recurrent, 3 primary tumors that later recurred and 4 recurrent tumors of HPV+ head and neck squamous cell carcinoma (HNSCCs). HPV+ HNSCCs in general have better prognosis; however, a fraction of these patients develop recurrence and have much worse prognosis. Hence, this study addresses a highly important question in the field, namely, what are the differences between recurrent and non-recurrent HPV+ HNSCCs. The authors found that the pathways related to TME remodeling (ECM, complement etc.) show significant differences. Moreover, recurrent tumors show defects in T cell functions and TCR signaling. The study will provide interesting and new information to the field of HPV+ HNSCCs and help to further elucidate the mechanisms of recurrence and define predictive markers.

Point-by-point comments:

1) In Figure 5, the significant proteins were shown with $p < 0.1$ between the NR and R groups. What if $p < 0.05$, any protein expression will be significantly different between NR and R?

Response: In the heatmaps in Fig. 5, we now added an asterisk (*) beside the protein name for the significant proteins with $p < 0.05$. Approximately a half of the kinases or phosphosites shown in the heatmaps have $p < 0.05$ between the NR and R groups. Moreover, all kinases or phosphosites in the heatmaps are with >2 -fold difference between the NR and R groups as indicated in the legend to Fig. 5.

2) It is unclear why only a few genes (PROS1, ANXA3, COL2A1, F3, TUBB3 and CD2) were chosen for predictive marker studies on page 11.

Response: A predictive marker was chosen when the following three conditions were met: 1) the protein abundance was significantly different between the no recurrence and recurrence groups, 2) the protein is a target of FDA-approved drug (Fig. 6, A and B), and 3) the expression significantly affects survival in the TCGA cohort. There are two issues here when comparing the proteomic data with the TCGA data. Firstly, the proteomic analysis by mass spectrometry quantifies protein abundance in the samples, whereas the TCGA data are transcriptomic data. It

has been reported that there is limited correlation between protein and RNA abundance. For example, Huang et al. (Cancer Cell, 2021, 39, 361–379) reported a low protein-RNA correlation for the complement system. Secondly, we are comparing recurrence from the proteome data with disease-specific survival from the TCGA cohort. This is because TCGA database has limited information about patient recurrence, and thus we used the survival as a sort of proxy to recurrence. Because of these limitations, we were able to identify only eight proteins that met all the conditions.

Moreover, these genes were chosen to represent the major functional modules/signaling networks involved in recurrence – fibrosis (COL2A1, TUBB3), immune suppression (CD2) and co-agulation (PROS1, ANXA3, F3). In the revised manuscript, we have added two more markers, PSMB5 and SLC7A11/xCT (pSer26) in Figure 7. SLC7A11 pSer26 site is a substrate of the kinase complex mTORC2, whose activation has been implicated in HNSCC disease progression (Ruicci, et al. Molecular Oncology 13 (2019) 2160–2177).

3) *Discussion is overly long and often repeated result description, which can be shortened.*

Response: We have removed redundancy and shortened the Discussion as suggested.

4) As the authors pointed out, the sample size is limited. However, the results are of great interest and should provide new information for the field to better elucidate the recurrence mechanisms.

Response: We appreciate this comment from the reviewer. As pointed out by Reviewer #1, recurrence samples are difficult to procure, especially for a relatively small cancer centre like ours.

Minor issues:

1) *There are some typos such as page 11, line 13, “Because aberrant regulation of these processed underlie recurrence of HPV+ HNSCC...”*

Response: We have checked the manuscript for typos and corrected this and other typos in the revision.

Reviewer #3 (Remarks to the Author):

The manuscript by Kaneko et al., entitled “Proteome and phosphoproteome signatures of recurrence for HPV+ head and neck squamous cell carcinoma” characterized recurrent and non-recurrent HPV+ HNSCC using MS-based proteomic and phosphoproteomic analysis. The authors have found significant differences related tumor microenvironment and epithelial-mesenchymal transition. Experimental design/method was well described and data analysis was carried out carefully. I have some concerns on the current manuscript that the authors should address.

1. *The author employed phosphoproteomic approach to characterize clinical cancer samples. Phosphorylation is sensitive to environmental changes, so it is required to explain the detail of cryopreservation after sample resection and the cryopreservation method.*

Response: The tumor samples were cryopreserved immediately after surgical resection using the optimal cutting temperature compound as cryo embedding matrix. We have now added the detail of cryopreservation in the Methods section.

2. *For sample quality check, the author mentioned that “Frozen section analysis was carried out to confirm tumor cellularity greater than 70%.” However, NR2 sample is localized in the control sample area in PCA plot Fig1A. From this figure, percentage of cancer cells in the NR2 sample is low. This discrepancy needs to be explained.*

Response: How the samples may be grouped on a plot depends on the clustering or dimensionality reduction method employed. In Figure 1B, we replaced the PCA plot with a UMAP (Uniform Manifold Approximation and Projection) plot that provides an alternative view. With this plot, NR2 can be seen as a part of the no recurrence cluster, although it is also close to the control cluster.

3. *Boosting technology was used to enhance pY identification. It is very powerful to increase identification number, however, concerned that over boosting by more than a factor of 100 will result in a loss of quantifiability. Therefore, it is recommended to exclude data where the intensity of the boost sample is extremely strong compared to the intensity of the sample.*

Response: We appreciate the reviewer’s concern with over-boosting for pTyr and have addressed it in the revised manuscript. Supplementary Table 6 lists pTyr sites with significant changes between the non-recurrent and recurrent tumours, based on the volcano plot in Fig. 1E. We have added a column that shows the average boost/sample intensity ratio in the linear scale for each pTyr site. For the 56 pTyr sites significantly different between the NR and R groups, the boost intensity ratio was >100-fold for 7 pTyr sites (shaded in grey in Table S6). Although we did not remove these data points from the volcano plot, we omitted description about these pTyr sites from the manuscript.

4. *Regarding EMT, were there differences in the expression levels of the typical markers CDH1 and CDH2?*

Response: We did not observe differences in either the expression or the phosphosite levels for CDH1/E-cadherin. CDH2 was not detected.

REVIEWERS' COMMENTS:

Reviewer #1 (Remarks to the Author):

The authors have addressed the concerns to the best of their ability. I believe the information contained is of interest to the scientific community and can provide meaningful foundational data to help understand the differences between HPV+ and negative cancers.

Reviewer #2 (Remarks to the Author):

I have no further comments. The authors have addressed all of my comments.

Reviewer #3 (Remarks to the Author):

None